# A Current Noise Cancellation Method Based on Fractional Linear Prediction for Bearing Fault Detection

**DOI:** 10.3390/s24010052

**Published:** 2023-12-21

**Authors:** Kaijin Xu, Xiangjin Song

**Affiliations:** School of Electrical and Information Engineering, Jiangsu University, Zhenjiang 212013, China; 2222207134@stmail.ujs.edu.cn

**Keywords:** fractional linear prediction (FLP), linear prediction (LP), time-shifting (TS), bearing fault detection, spectrum analysis

## Abstract

The stator current in an induction motor contains a large amount of information, which is unrelated to bearing faults. This information is considered as the noise component for the detection of bearing faults. When there is noise information in the current signal, it can affect the detection of motor bearing faults and lead to the possibility of false alarms. Therefore, to accomplish an effective bearing fault detection, all or some of these noise components must be properly eliminated. This paper proposes the use of fractional linear prediction (FLP) as a noise elimination method in bearing fault diagnosis, which makes these noise components the predictable components and this bearing fault information the unpredictable components. The basis of the FLP is to eliminate noise components in the current signal by predicting predictable components through linear prediction theory and optimal prediction order. Meanwhile, this paper adopts the FLP model with limited memory samples. After determining the optimal number of memories, only the fractional derivative order parameter needs to be optimized, which greatly reduces the computational complexity and difficulty in parameter optimization. In addition, this paper uses spectral analysis of the current signals through experimental simulation to compare the FLP method with the linear prediction (LP) method and the time-shifting (TS) method that have been successfully applied to bearing fault diagnosis. Based on the analysis results, the FLP method can better extract fault features and achieve better bearing fault diagnosis results, verifying the effectiveness and superiority of the FLP method in the field of bearing fault diagnosis. Additionally, the predictive performance of thevFLP and LP was compared based on experimental data, verifying the advantages of the FLP method in predictive performance, indicating that this method has a better noise cancellation effect.

## 1. Introduction

As significant energy conversion equipment, induction motors (IMs) are widely applied in agriculture, industry, and other fields. To guarantee the continuity of production and reduce the system’s maintenance costs, the health and stability of IMs are becoming increasingly important [1]. At present, the failure of the IM is the biggest factor for health and stability. Among the various fault types of the motor, there are bearing faults, stator winding faults, rotor faults, and others [2,3]. The majority (45–55%) of motor failures are related to bearings, including generalized roughness and single-point bearing faults [4,5]. When the bearing failure occurs, the damage to the bearing increases over time, which results in the motor being unable to operate properly [6]. Therefore, early bearing fault diagnosis is crucial for unnecessary downtime, reducing costs, and improving efficiency, and it has gradually become a research hotspot in the industry and academia [7,8].

Currently, the bearing failure diagnosis mostly employs vibration signal analysis, because the vibration signals are non-invasive and can detect motor faults without shutting down the unit [9]. Nevertheless, this method incurs additional investment costs and maintenance efforts with additional vibration sensors. Moreover, in some applications, after the erection of the plant, it is difficult to install the vibration sensors on the machines. In contrast, the motor current signals are easily accessible, which can decrease the investment and monitoring costs effectively [10]. Therefore, the diagnosis of bearing faults based on current signal analysis has emerged as a popular research topic [1].

The foundation of motor current signal analysis (MCSA) is the evaluation of motor currents [11]. Online fault diagnosis can be achieved while reducing costs through MCSA [12]. The characteristic of this method is the use of simple current clamps or current transformers, as well as the ability to achieve fast and efficient signal processing [10,13]. Additionally, the use of current signal analysis has shown effective results in diagnosing electrical faults [14]. However, when it comes to diagnosing bearing faults, the low signal-to-noise ratio (SNR) contributes to a relatively higher false alarm rate. This is because the current signal contains not only the fundamental waveform but also harmonics and non-Gaussian noise, which to some extent, interfere with the diagnosis. To improve bearing fault diagnosis, it is necessary to accurately extract fault feature components from the noisy signal [15]. Therefore, significant research efforts are required to fully eliminate noise components and improve the SNR of the current signal [16].

To improve the signal-to-noise ratio and eliminate the noise components in current signals, researchers have developed various noise reduction techniques to enhance the performance of bearing fault diagnosis. Ma et al. [17] proposed a deep learning-based fault diagnosis method for unmanned aerial vehicles, which uses empirical mode decomposition (EMD) to reduce high-frequency noise in the signal. However, EMD has mode aliasing and endpoint effects, which seriously affect the decomposition results, the calculation is complex, and the running speed is slow [18]. Zhao et al. [19] proposed a deep residual contraction network suitable for strong noise environments. This network uses a soft threshold denoising operator as a denoising module to eliminate noise. However, the denoising module does not consider the multi-scale distribution pattern of the fault signal, resulting in low diagnostic accuracy [20]. Lin et al. [21] proposed a motor-bearing fault detection method based on Gaussian filter denoising, Hilbert transform envelope extraction, and a convolutional neural network, which improved the performance of fault diagnosis. However, the Gaussian filtering cannot effectively denoise non-Gaussian noise. Gao et al. [22] proposed a composite fault diagnosis method for rolling bearings based on parameter-optimized maximum correlation kurtosis deconvolution (MCKD) and a convolutional neural network (CNN). The use of MCKD for signal denoising has improved the accuracy of fault diagnosis. However, these bearing fault diagnosis methods that use noise reduction are all based on deep learning techniques. The current deep learning technology still faces problems such as scarce fault data, imbalanced samples, and a long network training time [23].

Therefore, in response to the shortcomings of deep learning technology, noise suppression technology is adopted for the original signal to improve the SNR of the signal, thereby enhancing the fault feature extraction ability of the signal. By quickly identifying fault features through spectrograms, good bearing fault diagnosis results are achieved. Xu et al. [24] proposed the use of adaptive filtering algorithms, such as the least-mean-square (LMS) algorithm, for noise reduction. However, for traditional LMS algorithms, their steady-state bias increases rapidly with a fast convergence rate. This results in a lower signal-to-noise ratio and complicates the noise components, particularly in the presence of strong background noise. Zhou et al. [25,26] proposed an analog notch filter and a Wiener filter to estimate the noise components of most current signals. However, this approach requires re-optimization of the Wiener filter coefficients when changing the corresponding motor operating conditions. Additionally, it does not provide accurate noise prediction when the motor experiences other types of faults. In [27], an improved drive algorithm was proposed, which eliminates the power supply fundamental frequency and its second harmonic before data acquisition. This approach demonstrates its advantages, in low-speed detection particularly. Based on this algorithm, Pecht et al. proposed a current noise elimination method based on time-shifting (TS), which involves adding a delayed signal to the collected current signal as an anti-noise component to eliminate power supply fundamental frequency and odd harmonics. Unlike other methods, this method does not require the estimation of pure noise components, and the delay is only dependent on the power frequency and sampling rate [10]. Subsequently, Nazari et al. proposed a method of time-shifting in two phases of the three-phase current and integrating them with the third-phase current as a noise cancellation method, and the fault diagnosis is carried out through the spectrum of the residual square of synchronous current [28]. However, the method of reference [10] primarily focuses on eliminating the power supply fundamental frequency and odd harmonics, and the method of reference [28] primarily focuses on the fault of the single-point. To address other components such as eccentric harmonics, slot harmonics, and environmental noise in the stator current, Pecht et al. further proposed a noise suppression strategy using the linear prediction (LP) method [29]. This method separates the current signal into predictable and unpredictable components based on the characteristics of stator current signals related to bearing faults. The predictable component is modeled as the noise using optimal linear theory to eliminate all noise components from the current signal. Furthermore, the LP-based current noise elimination method can diagnose single-point bearing faults and generalized-roughness bearing faults. Principal component analysis (PCA) and wavelet analysis are also methods used for feature extraction and noise reduction. Sophian et al. [30] proposed a pulse eddy current (PEC) non-destructive testing feature extraction method based on PCA, which extracts relevant features through dimensionality reduction to achieve fault classification. Xu et al. [31] first denoised the original signal using wavelet thresholding, and then obtained the required feature parameters through PCA dimensionality reduction for the subsequent fault diagnosis. However, the selection of wavelet functions and thresholds in wavelet analysis is a challenge, and using the PCA method for dimensionality reduction may lose some information, which is usually used in conjunction with deep learning techniques [32]. The LP method directly predicts the original signal and its ability to eliminate noise is related to its predictive performance, without any loss of information. However, the use of the LP method involves relatively complex computations. As the order of LP increases, the corresponding prediction coefficients also increase, which leads to difficulties in parameter optimization.

To achieve effective current noise elimination while simplifying the computational complexity of the algorithm, this paper uses fractional linear prediction (FLP) as a noise reduction technique for bearing fault diagnosis. The FLP method offers lower computational complexity compared to the LP method while providing comparable predictive performance [33]. To demonstrate the effectiveness of the FLP algorithm in the field of bearing fault diagnosis, this paper conducted experiments to compare the fault feature extraction effectiveness of the TS and the LP algorithms, as well as the FLP algorithm. The results confirmed that the FLP method is effective in the field of bearing fault diagnosis and performs better than the TS and the LP algorithms. Based on experimental data, FLP has better predictive performance than LP, which means that FLP can better eliminate noise and ultimately achieve better fault feature extraction. Applying this method in the field of speech signals to the field of bearing fault diagnosis and achieving better results is a good innovation.

The structure of this paper is as follows. Section 2 provides an introduction to the relevant concepts of bearing faults and the theoretical background of the TS, LP, and FLP. Section 3 presents the specific methodology of using the FLP method for bearing fault diagnosis and selects the optimal FLP model for subsequent experimental validation. Section 4 describes the experimental setup, parameter configuration, and corresponding analysis of simulation results. Section 5 concludes this paper.

## 2. Preliminaries

In this section, the relevant theories of bearing faults are introduced. Subsequently, the implementation of the TS and the LP algorithms for bearing fault diagnosis and analysis are discussed. Finally, the theoretical knowledge of the FLP theory, which is the primary focus of this study, is presented.

### 2.1. Bearing Fault

Bearings are one of the fundamental mechanical components in motors, and their failures can result in the majority of machine failures and increase costs for troubleshooting and maintenance [34]. According to the stages in the fault development process, bearing faults can be classified into two types: generalized roughness and single-point bearing faults. The single-point bearing failure includes an inner ring failure, outer ring failure, ball failure, and cage failure [5]. This paper focuses on the outer ring fault in single-point faults. The corresponding theoretical characteristic frequencies of these bearing faults are as follows:(1)$$f_{i}=0.5\cdot N_{B}\cdot f_{r}\cdot \left(1+\frac{D_{b}\cdot \cos\theta }{D_{c}}\right)$$
(2)$$f_{o}=0.5\cdot N_{B}\cdot f_{r}\cdot \left(1-\frac{D_{b}\cdot \cos\theta }{D_{c}}\right)$$
(3)$$f_{b}=0.5\cdot f_{r}\cdot \left(\frac{D_{c}}{D_{b}}\right)\cdot \left[1-{\left(\frac{D_{b}\cdot \cos\theta }{D_{c}}\right)}^{2}\right]$$
(4)$$f_{c}=0.5\cdot f_{r}\cdot \left(1-\frac{D_{b}\cdot \cos\theta }{D_{c}}\right)$$
where *f_i_*, *f_o_*, *f_b_*, and *f_c_* are the characteristic frequencies of the inner-race bearing fault, outer-race bearing fault, bearing ball fault, and bearing cage fault, respectively, *f_r_* is the rotating frequency of the bearing, and *N_B_* is the number of balls in the bearing. *D_b_* is the ball diameter, *D_c_* is the pitch diameter, and *θ* is the ball contact angle, which is normally zero [35,36].

The frequency of the sideband fault characteristic is typically considered when performing spectrum analysis on current signals:(5)$$\left|f_{1}\pm f_{f}\right|$$
where *f*_1_ represents the fundamental frequency of the power source, and *f_f_* represents one of the four-fault frequencies mentioned above.

In the simulation verification of the following three main methods, the frequency of the sideband fault characteristic is an important basis for detecting whether bearings have faults.

### 2.2. Time-Shifting

In the time-shifting method, the primary components of the stator current are divided into two parts: one part consists of the fundamental harmonic and odd multiple harmonics from the power source, while the other part includes the remaining current components. The core of this method is to generate delayed current signals by adding a delay and then subtracting them from the original current signals to obtain residual current signals, which are subsequently subjected to frequency spectrum analysis for fault diagnosis. The fault information in the residual current signals is easier to identify because the non-fault components and the noise have been eliminated through the subtraction process [10].

The time-shifting method is used to represent the supply fundamental and its odd multiple harmonics as an amount of *N*, resulting in the delayed current signals *i_delayed_*. Here, *N* is a numerical representation devised for convenience in development. In practice, the time-shifting method introduces a delay of a specific time interval, which is *t* and is equal to $t=\frac{\pi }{\omega }$. The delayed current is then added to the current component *i_F_* and finally the residual current signal, which is further subjected to spectral analysis and fault characteristic diagnosis analysis; this approach is illustrated in Figure 1.

*i_F_* represents the stator current, and ω represents the power source angular frequency. The formula corresponding to *N* is as follows:(6)$$N=\frac{F_{S}}{2f_{1}}$$
where *F_s_* is the sampling frequency of the current signal, and *f*_1_ is the frequency of the power supply.

However, this method only eliminates the fundamental and odd harmonics of the power supply and cannot eliminate eccentricity and other harmonic components. It can only be applied to single-point bearing faults and cannot diagnose generalized-roughness bearing faults [22].

### 2.3. Linear Prediction

To address the drawbacks of the TS method, the LP linear prediction theory was proposed, which is not only applied in the field of digital signal processing but also serves as a noise elimination method in bearing fault diagnosis. The model used for LP is autoregressive and can be described by the following equation [22]:(7)$$y\left[n\right]=\sum_{k=1}^{P}\alpha_{k}x\left[n-k\right]$$
where *y*[*n*] represents the current predicted signal, *α_k_* represents the prediction coefficient, and *P* represents the prediction order. The variable *x* represents the measured signal, and the entire equation represents how the current predicted value *y* is obtained by taking a weighted sum of *P’s* previously measured signal values.

In the stator current signal, the current fundamental frequency and its various harmonic components are defined as the predictable components, while the bearing fault signal and the background Gaussian noise are defined as the unpredictable components. The basis of applying the LP method is to eliminate the predictable components and define them as the noise components [22].

The LP method is used to predict the predictable signals *i_predicted_*. Then, the predictable components *i_predictable_* and the unpredictable components *i_unpredictable_* of the original current signals *i_F_* are subtracted from the predicted signals *i_predicted_* obtained through the use of the LP linear prediction method. The prediction order *P* represents the number of samples used for prediction. Determining an optimal prediction order can result in better cancellation of predictable components and a better effect of noise elimination. The resulting residual signal is then subjected to frequency spectrum analysis to reveal the fault characteristic frequencies, which are used for the bearing fault diagnosis analysis, as illustrated in Figure 2.

In the subsequent experimental analysis, the calculation formula for the prediction order *P* in the LP method is the same as the calculation method for the *N* in Equation (6) [22].

The LP method is an improved method of the TS method, which can better eliminate noise components in signals. This algorithm has good predictive performance, but its computational complexity and parameter optimization are relatively complex. Therefore, the FLP method mainly studied in this article is used to solve this problem, and the following section also first introduces the theory of the FLP method.

### 2.4. Fractional Linear Prediction

The FLP is a modern method for LP that has been adopted in various fields for prediction purposes [33,37]. It can be expressed as a linear combination of *q* “fractional terms” and written as follows:(8)$$\dot{x}[n]=\sum_{i=1}^{q}aD^{\alpha }x[n]$$
where *ẋ*[*n*] is the predicted signal, *a* is the FLP coefficient, *α* is the order of the fractional derivative, and *D^α^* is the fractional derivative that can be interpreted using the Grünwald–Letnikov(GL) definition, which is widely used for the numerical solution of the fractional derivative of a function *x*(*t*) at time instant *t*:(9)Dtαaxt=limh→01hα∑j=0k=t−ah−1jαjxt−jh
where *K* is the integer part of the fraction (*t − a*)/*h*, *a* and *t* are lower and upper terminals of differentiation, respectively, and *α* is the arbitrary real order of the fractional derivative. So, first of all, we put *t = nh* into (9) and then denote the *n*-th sample of the signal *x*(*nh*) as *x*[*n*], i.e., *x*[*n*] = *x*(*nh*), and (10) can be rewritten as follows:(10)Dtαaxt=limh→01hα∑j=0k=t−ah−1jαjxt−jh

The FLP model used in this article is a restricted memory optimal FLP model. Only the first term in Equation (8) is utilized in this model, where *D^α^x*[*n* − 1] is shifted back in time by one sample, and this can be written as follows:(11)$$\dot{x}[n]=aD^{\alpha }x[n-1]$$

Then, assuming an effective positive integer value “*m*” as the upper limit for the summation in Equation (10), Equation (12) can be obtained [33]:(12)$$D^{\alpha }x[n-1]=\frac{1}{h^{\alpha }}\sum_{j=0}^{m}{\left(-1\right)}^{j}\left(\begin{array}{c} \alpha \\ j \end{array}\right)x[n-1-j]$$

By combining Equations (11) and (12), the current signal *ẋ*[n] from the prediction can be obtained, which can be written as follows:(13)$$\dot{x}[n]=\frac{a}{h^{\alpha }}\sum_{j=0}^{m}{\left(-1\right)}^{j}\left(\begin{array}{c} \alpha \\ j \end{array}\right)x[n-1-j]$$

From Equation (13), it can be seen that when the number of memories *m* of the FLP model with limited memory samples is determined, the model only has one coefficient to be optimized, which is the parameter *a*. This coefficient can also be represented by the order of the fractional derivative *α* [33,38]. Therefore, in the experiment, only the optimal order of the fractional derivative needs to be set, and in the FLP model with limited memory samples, there are only three cases where the number of samples is limited. As mentioned in reference [33], the optimal order is the reciprocal of the number of samples, which results in the best prediction performance, greatly reducing the difficulty of parameter optimization. Although the LP method only needs to calculate the optimal prediction order in bearing fault diagnosis, according to Equation (7), the predicted signal is a weighted sum of *P* coefficients containing prediction coefficient *a*. According to Equation (6), assuming a sampling rate of 51.2 KHz and a power supply fundamental frequency of 50 Hz, the predicted order *P* is calculated as 512. However, optimizing 512 prediction coefficients significantly increases the difficulty of the optimization process. This significantly reduces the computational complexity and is one of the advantages of the FLP method.

From the analysis of the relevant literature and the previous paragraph, it has been demonstrated that the computational complexity of the LP method is higher than that of the FLP method. Moreover, the predictive performance of the FLP method is comparable to that of the LP method, thus showing that FLP, as a predictive algorithm, has certain advantages. Furthermore, a fully stored FLP model has been proposed, which can be used for complete signal history [38]. In the next section, the method FLP for noise elimination and the bearing fault diagnosis are described. Then, a comparison is made between the two different FLP models to select the optimal one for subsequent experimental verification and comparison.

## 3. Current Noise Elimination in Bearing Fault Detection Based on FLP

### 3.1. Bearing Fault Detection Based on FLP

Based on the previous analysis, the FLP algorithm is applied to diagnose the bearing fault and eliminate the noise components in the current signal. This approach is employed to effectively extract characteristic components related to bearing faults and facilitate the analysis of bearing fault diagnosis. The framework of this method mainly consists of data acquisition, algorithm processing, and spectral analysis.

First, the stator current of a single phase in a three-phase motor is acquired through a data acquisition system to obtain the required current signal data. Similar to the LP method, the current component is divided into predictable and unpredictable components. The predictable components represent the noise that needs to be eliminated. However, in the FLP method, linear prediction involves utilizing the number of memory storage units denoted by *m* used as the count of predictor terms. Similar to *P* in the LP method, the introduction of the number of memory units, denoted as *m*, is used to predict the noise signal. For FLP models with restricted memory samples, only three values of *m* are utilized: 1, 2, and 3. Therefore, the corresponding count of memory samples is defined as the parameter *M*, which are 2, 3, and 4, respectively. On the other hand, for the FLP model with full memory, there is no restriction on the value of *m*. The predicted results are then subtracted from the two components of the original signal to obtain the residual signal. A frequency spectrum analysis of the residual signal is carried out to extract fault characteristics, as illustrated in Figure 3.

### 3.2. Determination of the FLP Model

Before the next section of the experimental comparative analysis, it is necessary to determine the FLP model, including selecting the optimal value of *m* if choosing an FLP model with memory samples. The criterion for selection is the prediction performance of the FLP model under the corresponding signal. The prediction gain (PG) in dB is used as the metric. The signal in this case is a standard sinusoidal waveform, which is denoted as follows:$$x\left(t\right)=\sin(2\pi t)$$

First, the prediction performance of the FLP model with restricted memory samples is compared, where the values of *M* are set to 2, 3, and 4. The corresponding fractional derivative order *α* is set as the reciprocal of the memory samples, which represents the optimal fractional derivative order [33].

From the simulation results corresponding to the three different memory sample sizes, it can be observed that evaluating their prediction performance directly from the images is challenging, as illustrated in Figure 4. However, the calculated results of the prediction gain still provide a distinguishable comparison.

Table 1 shows that under the corresponding optimal fractional derivative order, when the number of memory samples *M* is 2, the value of prediction gain is the highest, indicating the best predictive performance. Further optimization of the value of *α* resulted in the calculation of prediction gain once again.

Table 2 reveals that by adjusting the order magnitude, the prediction gain results of FLP models with three different memory samples have been improved. However, the highest numerical value is still observed when the number of memory samples *M* is 2.

To verify that the prediction performance of the FLP model with limited memory samples is optimal when the number of storage units M is 1, the comparison of simulation signals is supplemented by analyzing the prediction gain of the current data collected in subsequent experiments.

Table 3, Table 4 and Table 5, respectively, provide a comparison of predicted gains for three sets of current data: light load, medium load, and heavy load. From the three tables, it can be seen that when the number of memory samples is 2, the prediction gain is the highest. Therefore, when the number of memory samples M is set to 2 and the number of storage units m is 1, the predictive performance in FLP models with restricted memory samples is optimal.

Next, the same sine wave signal is used for the simulation analysis on the full-memory FLP model. Unlike the FLP model with restricted memory samples, the full-memory FLP model is not constrained by a finite number of memory samples and is essentially similar to the LP model [38]. The results show that although the maximum achievable prediction gain value is comparable to that of the FLP model with restricted memory samples, optimization becomes complex when comparing with LP methods that have higher prediction order settings, as the corresponding order of the full-memory FLP model needs to be set with the same numerical values.

In conclusion, considering the difficulty in parameter optimization, the FLP model with restricted memory samples is superior to the FLP model with full memory. For the upcoming comparative analysis in the next section, the FLP model with restricted memory samples, specifically with *M* = 2 memory samples and *m* = 1 storage units, is chosen. In the following section, the experimental settings, as well as the fault feature extraction performance of TS, LP, and FLP, are analyzed and compared.

## 4. Experimental Design and Result Analysis

In this section, the experimental setup and collected data are introduced. Based on the collected relevant data, the feasibility of the proposed FLP method in the field of bearing fault diagnosis as well as its superiority compared with the original current, TS method, and LP method are simulated and analyzed. The results are to be used to make conclusions.

### 4.1. Experimental Setup and Preparation before Analysis

The experimental setup utilized in this study consists of three parts: a power system, a motor driving system, and a data acquisition system. The power system is supplied with electricity from a 50 Hz AC power source. The motor driving system comprises an experimental motor and a Z2-42 DC generator, which are used to adjust the experimental motor speed by varying the excitation voltage of the DC generator. The data acquisition system consists of a JLB-21 current transformer, a self-made conditioning circuit, and an NI data acquisition card. The experimental motor used in the study is a Y100L2-4 three-phase asynchronous motor with a rated power of 3 kW, rated voltage of 380 V, and rated current of 6.8 A. The degree of failure of the bearing outer ring is severe. The experimental platform for the collection of motor data and the bearing with outer ring fault used in the experiment are shown in Figure 5.

During the experiment, the asynchronous motor is operated at different load conditions by setting different excitation voltages for the DC generator. First, the stator current data of the motor with an outer ring fault in the bearing are collected and followed by data collection for the motor without any faults. During the data collection of the motor with an outer ring fault in the bearing, three different types of data were collected: light load, medium load, and heavy load. For each load type, two sets of data are collected, resulting in a total of six data sets ranging from light load to heavy load. These sets of data are numbered 1–6 accordingly. All collected data are sampled at a rate of 25 KHz. Finally, the corresponding spectral analysis is performed on these six sets of data.

To validate the effectiveness of the FLP method, this study analyzes the current data of a motor with an outer race-bearing fault under six different load conditions. The main focus is to compare the results of the FFT spectrum analysis between the raw current and the residual current processed by the TS, LP, and FLP algorithms. Table 6 presents the theoretically calculated values of the fault characteristic frequencies of the outer race bearing under six load conditions, as well as the corresponding sideband fault characteristic frequencies.

Before the commencement of simulation analysis, the three methods in this study are first configured with appropriate parameters. According to Equation (6), the delay amount for TS is set to the corresponding calculated value of 250. Based on the previous theoretical discussion, the prediction order for LP is also set to the same value of 250. Additionally, the FLP model used is the one validated in the previous section with a limited sample memory of *m* = 2. Finally, the order of fractional differentiation *α* is also similarly adjusted to 0.001.

### 4.2. FFT Spectrum Analysis of Current Signals

#### 4.2.1. Comparison of Frequency Spectrum Analysis between Healthy Motor and Faulty Motor

Before comparing the four methods for the faulty motor with an outer ring bearing fault, it is necessary first to compare the spectrum analysis of a healthy motor and a faulty motor under a heavy load. Figure 6 presents the FFT frequency spectrum analysis (0–150 Hz) of a healthy motor under a heavy load. The presence of the inherent eccentric harmonic component *f*_1_ ± *f*_r_, caused by the motor’s internal eccentricity, makes it difficult to observe the sideband fault frequencies.

However, when the motor experiences outer-ring bearing faults, as shown in Figure 7, the characteristic frequencies associated with eccentricity faults become more prominent. In this case, while the sideband fault frequency |*f*_1_
*– f_of_*| may not be visible, *f*_1_ + *f_of_* can still be recognized. Based on the experimental findings, it is evident that the FLP method has a significant capability to eliminate noise. The interference component at the observable peaks corresponding to eccentricity harmonics is relatively reduced compared to the interference observed in the other two methods. In other words, the FLP method can effectively eliminate noise components and suppress irrelevant components. Only the fault characteristic frequency or harmonic components to be analyzed will be displayed on the graph. As a result, the FLP method exhibits a more significant and distinct efficacy in extracting the fault characteristic frequency and eccentric fault frequency.

To further validate the effectiveness of the FLP method, a comparative simulation of FFT frequency spectrum analysis is conducted on the original current and the residual current of TS, LP, and FLP methods for motors with outer-ring bearing faults under six different loads. The purpose is to compare the fault characteristic extraction performance of the four methods and verify whether the FLP method is the best. The frequency range is set between 130 Hz and 150 Hz.

#### 4.2.2. Comparison and Analysis of Frequency Spectrum Analysis for Outer-Ring Bearing Faults under Light Load Conditions

The frequency spectrum analysis of a faulty motor with the outer-ring bearing fault under light load conditions is shown in Figure 8. Under the light load conditions, it is noticeable that the performance of the frequency spectrum analysis using the raw current signal is observed, and the extraction of sideband fault characteristic frequencies is not effective due to the interference of noise. However, when analyzing the residual current using the TS, LP, and FLP methods, the extraction of sideband fault characteristic frequencies is improved, and the spectral line *f*_1_ + *f_of_* is visible. It can be observed from both Figure 8a,b, that the FLP method exhibits the most prominent spectral line *f*_1_ + *f_of_*, indicating that it is the most effective in extracting sideband fault characteristic frequencies. Additionally, it can be observed that the FLP method significantly reduces the overall noise level, with the lowest amplitude. Normally, under low load conditions, the performance of fault feature extraction is lower than under heavy load conditions. However, from the experimental results, it can be observed that the effect of bearing fault feature extraction using the FLP method is still very obvious compared to other methods. This also reflects the superiority of the FLP method.

#### 4.2.3. Comparative Analysis of Frequency Spectrum Analysis for Outer-Ring Bearing Faults under Medium and Heavy Load Conditions

The frequency spectrum analysis results for a faulty motor with the outer-ring bearing fault under medium and heavy load conditions are depicted in Figure 9.

From the experimental results, it can be observed that when performing frequency spectrum analysis using the raw stator current signal, the spectral line *f*_1_ + *f_of_* is distinguishable, and its extraction effectiveness improves as the load increases. Similarly, for the other three methods, the spectral line *f*_1_ + *f_of_* remains visible. The FLP method continues to exhibit the best extraction of corresponding sideband fault characteristic frequencies, and the overall noise level is also minimized.

In addition, a comparison of predictive performance is made for the current data under six load groups, using mean square error (MSE) as the evaluation criterion. From Table 7, it can be seen that the MSE value of FLP is much smaller than LP, which also confirms that FLP has better predictive performance than LP in this experiment.

By conducting an FFT spectrum analysis on current signals under three different types of loads, it can be observed that, compared to the other three methods, the use of the FLP method for linear prediction of stator current signals, followed by spectrum analysis of the resulting residual current signals, yields the best results in fault characteristic extraction. Moreover, as depicted in Figure 7, Figure 8 and Figure 9, when using the FLP algorithm, there is a significant reduction in overall noise amplitude, and the sideband fault characteristic frequencies are less affected by the noise. Additionally, the extraction of eccentric fault frequencies, apart from the outer-ring sideband fault characteristic frequencies, becomes clearer. Through the comparison of experimental data, it is found that FLP has better predictive performance than LP. These results validate the feasibility of applying the FLP algorithm to bearing fault diagnosis and demonstrate the superiority and effectiveness of the FLP algorithm in fault feature extraction and predictive performance compared to other algorithms proposed in the field of fault diagnosis.

## 5. Conclusions

FLP has been applied in previous studies for modeling and compression of speech signals due to its good predictive performance [33,37]. The LP method [22], which is used for predicting faults, has also been utilized in the area of bearing fault diagnosis. This predictability is used as a distinguishing feature, and the predictable components are modeled as noise to eliminate noise and highlight fault characteristics, achieving the goal of fault diagnosis. This paper proposes to apply the FLP method to the field of bearing fault diagnosis, achieving good bearing fault diagnosis results.

The core of FLP is to use the linear prediction theory to make the noise contained in the current signal the predictable components and achieve noise elimination by subtracting the predicted signal from the original current signal. Through spectrum analysis of residual current signals after noise elimination in the TS method, LP method, and FLP method, it can be found that the FLP method has the most obvious effect on extracting fault features, whether it is sideband fault frequency or eccentric fault frequency. Moreover, the noise elimination effect using the FLP method is particularly notable, resulting in a significant reduction in the overall amplitude and a clearer representation of fault frequencies. Based on experimental data, using MSE as the predictive performance evaluation criterion, the predictive performance of FLP is significantly better than LP, proving that this method has better predictive performance in bearing fault diagnosis, which means that the noise elimination effect is also better. FLP solves the problem of difficulty in optimizing LP parameters. These results also validate the effectiveness of the FLP method in bearing fault diagnosis and its advantages over other methods. Moreover, compared to deep learning techniques, the FLP method has a faster diagnostic speed, which can directly detect whether a fault has occurred from the spectrum graph. The application of predictability as a distinguishing feature eliminates the need to understand pure noise data under different conditions during the noise modeling process. Engineers can use the proposed methods for predictive maintenance, which can prevent missed and false alarms and prevent catastrophic failures.

Future work should apply this current noise elimination method FLP to other motor bearings to verify its generalization and whether it can also achieve good bearing fault diagnosis results and whether the parameters of the FLP model in this article are also used in different motor bearings. Further research is required for this work.

## Figures and Tables

**Figure 1 sensors-24-00052-f001:**
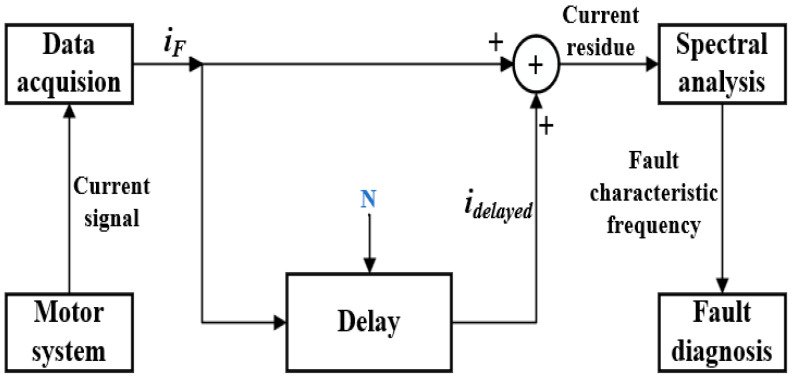
The principle framework of the time-shifting method.

**Figure 2 sensors-24-00052-f002:**
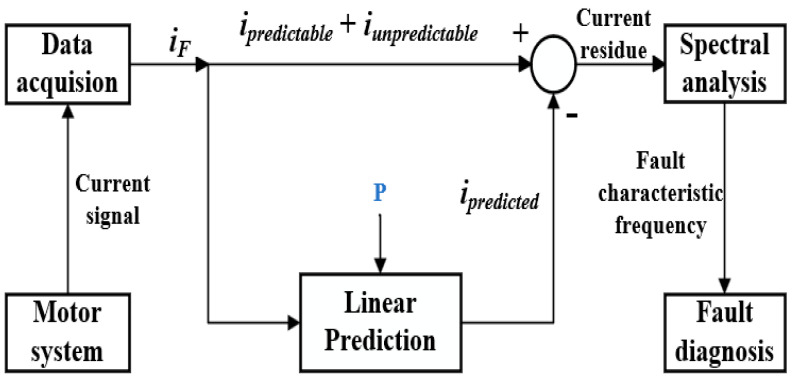
The principle framework of the LP method.

**Figure 3 sensors-24-00052-f003:**
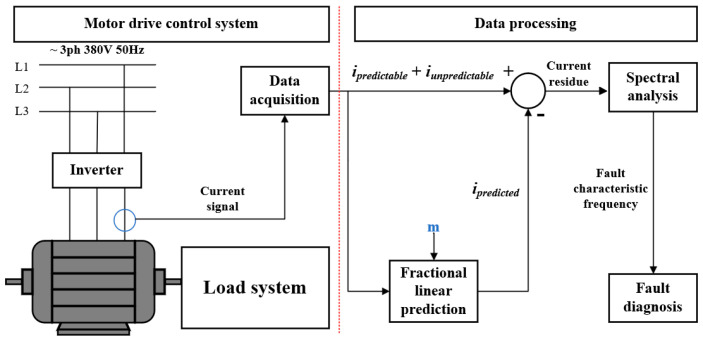
The principle framework of the FLP.

**Figure 4 sensors-24-00052-f004:**
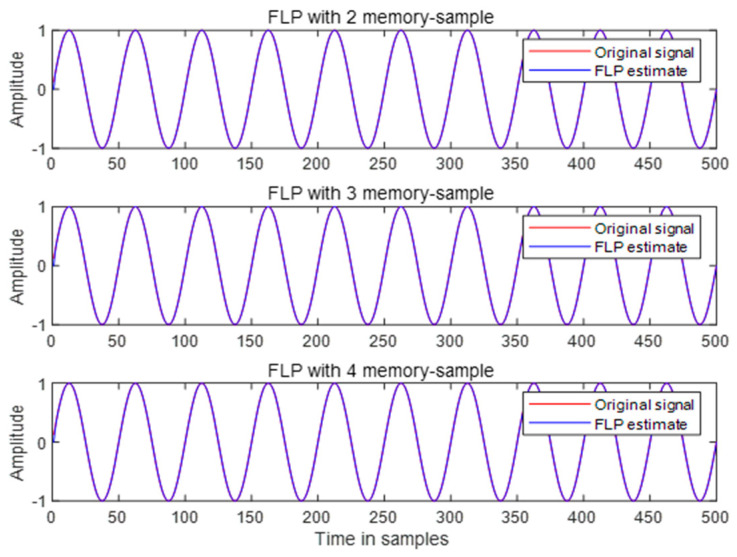
FLP with restricted memory.

**Figure 5 sensors-24-00052-f005:**
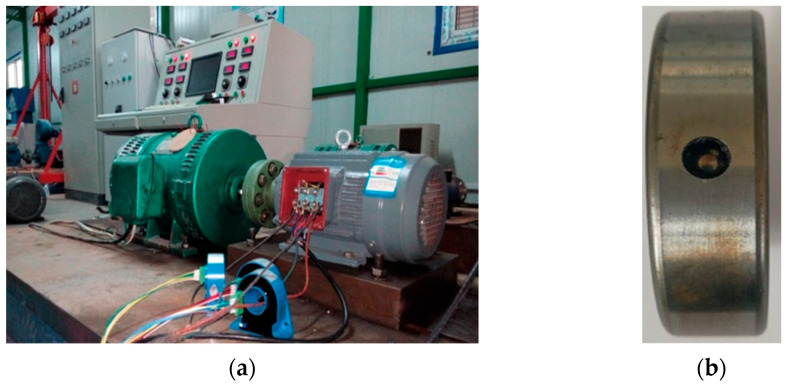
(**a**) Electric motor data acquisition experimental platform. (**b**) Outer-race fault bearing.

**Figure 6 sensors-24-00052-f006:**
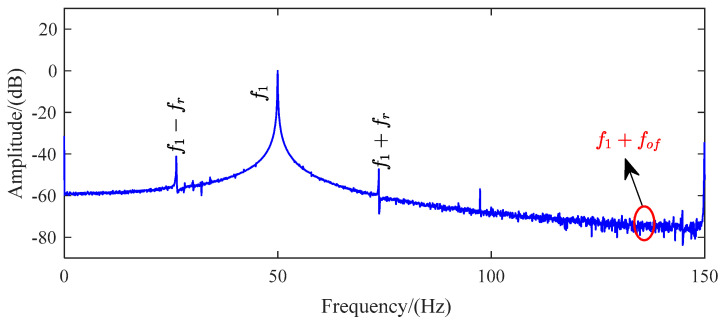
Spectrum analysis of healthy motor current.

**Figure 7 sensors-24-00052-f007:**
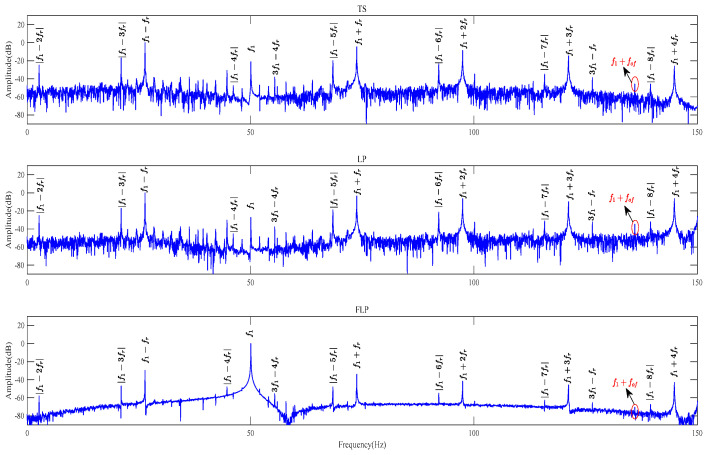
Frequency spectrum analysis of residual current for TS, LP, and FLP in faulty motors with outer-ring bearing faults.

**Figure 8 sensors-24-00052-f008:**
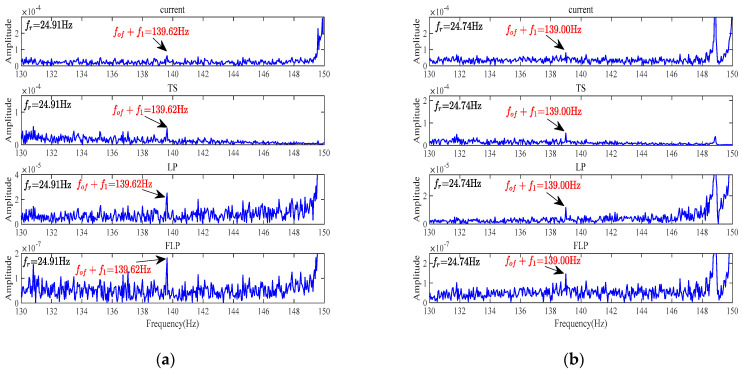
FFT spectrum analysis of faulty motor under light load conditions. (**a**) Faulty motor, load 1. (**b**) Faulty motor, load 2.

**Figure 9 sensors-24-00052-f009:**
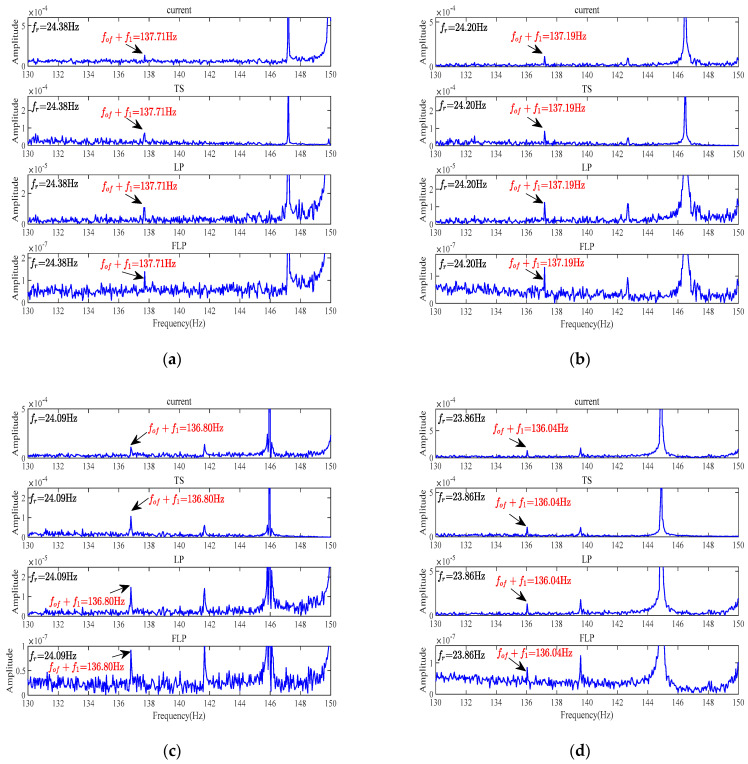
FFT spectrum analysis of faulty motor under medium and heavy load conditions. (**a**) Faulty motor, load 3. (**b**) Faulty motor, load 4. (**c**) Faulty motor, load 5. (**d**) Faulty motor, load 6.

**Table 1 sensors-24-00052-t001:** Predictive performance of sine signals given in units of PG [dB].

M	α	FLP
2	0.5	41.76
3	0.33	40.64
4	0.25	40.67

**Table 2 sensors-24-00052-t002:** Predictive performance of sine signals with improved order given in units of PG [dB].

M	α	FLP
2	0.5029–0.5051	42.00–42.02
3	0.3381–0.3385	41.85
4	0.2558–0.2564	41.64

**Table 3 sensors-24-00052-t003:** Predictive performance of light load experimental data given in units of PG [dB].

M	α	FLP
2	0.01	23.18
3	0.01	23.16
4	0.01	23.15

**Table 4 sensors-24-00052-t004:** Predictive performance of medium load experimental data given in units of PG [dB].

M	α	FLP
2	0.01	26.65
3	0.01	26.64
4	0.01	26.63

**Table 5 sensors-24-00052-t005:** Predictive performance of heavy load experimental data given in units of PG [dB].

M	α	FLP
2	0.01	28.94
3	0.01	28.93
4	0.01	28.93

**Table 6 sensors-24-00052-t006:** Theoretical fault sideband components of the healthy and faulty IM.

Load	Fundamental Frequency Estimation Value *f*_1_/Hz	Estimated Rotational Speed Value *f*_r_/Hz	Outer Race Fault Characteristic Frequency (*f*_of_ = 3.6 *f*_r_)/Hz	Sideband Fault Characteristic Frequencies |*f*_1_ – *f*_of_|/Hz and *f*_1_ + *f*_of_/Hz
1	49.97	24.90	89.67	39.69, 139.64
2	50.02	24.74	89.05	39.03, 139.07
3	49.92	24.38	87.78	37.86, 137.71
4	50.07	24.21	87.15	37.08, 137.21
5	50.07	24.09	86.73	36.67, 136.81
6	50.07	23.86	85.91	35.84, 135.97

**Table 7 sensors-24-00052-t007:** The performance of the predictions given in MSE.

Load	LP	FLP
1	6.8368 × 10^−5^	1.6108 × 10^−7^
2	5.5831 × 10^−5^	8.9460 × 10^−8^
3	6.2421 × 10^−5^	7.1168 × 10^−8^
4	6.2150 × 10^−5^	5.5823 × 10^−8^
5	6.0166 × 10^−5^	4.9200 × 10^−8^
6	6.2059 × 10^−5^	4.2108 × 10^−8^

## Data Availability

Data are contained within the article.

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
