# Peer review of "A Current Noise Cancellation Method Based on Fractional Linear Prediction for Bearing Fault Detection"

_sensors, 2023, doi:10.3390/s24010052_

Round 1

Reviewer 1 Report

Comments and Suggestions for Authors

The structure of the manuscript is relatively complete, and the proposed fault diagnosis method has a certain degree of innovation. The theoretical introduction of the compared methods is important, and the experimental verification effect is good. However, there are still some problems, which can be summarized as follows:

1.     It is only verified that the optimal number of storages is 1 in sine wave signals and it needs to be determined that this number is also optimal in subsequent experimental data.

2.     The introduction mentions that the predictive performance of FLP and LP is equivalent, and it is necessary to increase the comparison of their predictive performance on experimental data in subsequent experimental verification.

3.     For the complexity of parameter optimization, the comparison between FLP and LP is not specific enough.

4.     There are some grammar errors in the language of this manuscript, and some vocabulary usage is not appropriate enough, which needs to be modified.

Comments on the Quality of English Language

Minor editing of English language required

Author Response

Comments to the Author

The structure of the manuscript is relatively complete, and the proposed fault diagnosis method has a certain degree of innovation. The theoretical introduction of the compared methods is important, and the experimental verification effect is good. yet, some important modifications have to be done.

Comment 1. It is only verified that the optimal number of storages is 1 in sine wave signals and it needs to be determined that this number is also optimal in subsequent experimental data.

Response: We have added a comparison of the predictive performance of experimental data for the following three load scenarios. In Tables 3, 4, and 5 of the section 3, it can be found that when the number of memory samples is 2, the prediction gain PG value is the highest. This further confirms that the optimal storage quantity of the FLP model is 1, which is optimal.

Comment 2. The introduction mentions that the predictive performance of FLP and LP is equivalent, and it is necessary to increase the comparison of their predictive performance on experimental data in subsequent experimental verification.

Response: Thank you very much for the comment. At the end of the experimental section in the section 4, a comparison of the predictive performance of FLP and LP for the experimental data was added. In Table 7, it can be found that the MSE value was used as the evaluation criterion for the predictive performance of 6 sets of data, and FLP's MSE value was much smaller than LP's. This means that FLP method has better predictive performance in bearing fault diagnosis than LP, reflecting the advantages of FLP method in predictive performance.

Comment 3. For the complexity of parameter optimization, the comparison between FLP and LP is not specific enough.

Response: Thank you very much for the comment. We have added an explanation for the LP equation. Quantitative calculations were conducted based on LP expressions. According to equation (7) in the second section, the predicted signal is the weighted sum of P coefficients containing the prediction coefficient a. According to equation (6), assuming a sampling rate of 51.2KHz, a power supply fundamental frequency of 50Hz, and a predicted order P of 512. There are 512 prediction coefficients a that need to be optimized, greatly increasing the difficulty of optimization,which Verified the difficulty of parameter optimization in LP method compared to FLP.

Comment 4. There are some grammar errors in the language of this manuscript, and some vocabulary usage is not appropriate enough, which needs to be modified.

Response: Thank you very much for the comment. We apologize for the multiple grammar errors in this manuscript, which have reduced readability. The main issues are tense, singular and plural, and some definite articles. After reading the manuscript multiple times, these errors have been highlighted in blue in the manuscript.

Reviewer 2 Report

Comments and Suggestions for Authors

The work could be interesting. Significant improvement is required.

1. More research background of noise reduction and Bearing Fault Detection should be reviewed and discussed e.g.  Yumeng Ma, Faizal Mustapha, Mohamad Ridzwan Ishak, Sharafiz Abdul Rahim & Mazli Mustapha (2023) Structural fault diagnosis of UAV based on convolutional neural network and data processing technology, Nondestructive Testing and Evaluation, DOI: 10.1080/10589759.2023.2206655;

2. PCA and wavelet analysis for feature extraction and noise reduction could be discussed and compared e.g. Ali Sophian, etc., A feature extraction technique based on principal component analysis for pulsed Eddy current NDT, NDT & E International, Volume 36, Issue 1, 2003, Pages 37-41, ISSN 0963-8695, https://doi.org/10.1016/S0963-8695(02)00069-5.

3. Mind technical contribution and novelty;

4. The conclusion and future work should be enhanced.

Comments on the Quality of English Language

English proofread is expected.

Author Response

Comments to the Author

The work could be interesting. Significant improvement is required.  

Comment 1. More research background of noise reduction and Bearing Fault Detection should be reviewed and discussed e.g. Yumeng Ma, Faizal Mustapha, Mohamad Ridzwan Ishak, Sharafiz Abdul Rahim & Mazli Mustapha (2023) Structural fault diagnosis of UAV based on convolutional neural network and data processing technology, Nondestructive Testing and Evaluation, DOI: 10.1080/10589759.2023.2206655.

Response: Thank you very much for the comment. We have added research on noise reduction methods and bearing fault diagnosis in the introduction. Firstly, the example you provided is analyzed. The fault diagnosis method in this paper was based on deep learning technology and EMD method was used to denoise the signal. However, EMD has mode aliasing and endpoint effects, which seriously affect the decomposition results, and the calculation is complex and the running speed is slow. Zhao et al. proposed a deep residual contraction network suitable for strong noise environments. This network uses a soft threshold denoising operator as a denoising module to eliminate noise. However, the denoising module did not take into account the multi-scale distribution pattern of the fault signal, resulting in low diagnostic accuracy. Lin et al. proposed a motor bearing fault detection method based on Gaussian filter denoising, Hilbert transform envelope ex-traction, and convolutional neural network, which improved the performance of fault diagnosis. However, Gaussian filtering cannot effectively denoise non-Gaussian noise. Gao et al. proposed a composite fault diagnosis method for rolling bearings based on parameter optimized maximum correlation kurtosis deconvolution (MCKD) and convolutional neural network (CNN). The use of MCKD for signal denoising has im-proved the accuracy of fault diagnosis. However, these bearing fault diagnosis methods that use noise reduction are all based on deep learning techniques. The current deep learning technology still faces problems such as scarce fault data, imbalanced samples, and long network training time. Due to the shortcomings of the fault diagnosis method based on the combination of deep learning and denoising methods, subsequent denoising methods are introduced to improve the effectiveness of fault feature extraction through denoising. It is possible to visually determine whether a bearing fault has occurred from the frequency spectrum. The added references are as follows:

[1] Ma, Y.M.; Mustapha, F.; Ishak, M.R.; Rahim, S.A.; Mustapha, M. Structural fault diagnosis of UAV based on convolutional neural network and data processing technology. Nondestructive. Test. Eva. 2023, 1-20.

[2] Yang, L.; Cao, L.; Wang, J.L.; Yao, X.H.; S, Y.; Wu, Y.J. Bearing fault feature extraction measure using multi-layer noise reduction technology. In Proceedings of the 2022 IEEE International Conference on Sensing, Diagnostics, Prognostics, and Control ( SDPC), Chongqing, China, 5-7 August 2022; pp. 53-56.

[3] Zhao, M.; Zhong, S.; Fu, X.; Tang, B.; Pecht, M. Deep residual shrinkage networks for fault diagnosis. IEEE T. Ind. Informat. 2020, 16, 4681-4690.

[4] Wang, X.; Zhang, H.; Du, Z. Multiscale Noise Reduction Attention Network for Aeroengine Bearing Fault Diagnosis. IEEE T. Instrum. Meas. 2023, 72, 1-10.

[5] Lin, A.; Run, Y.; Zhao, W. A Motor Bearing Fault Diagnosis Method Based on Signal Analysis and Convolutional Neural Network. In Proceedings of the 2022 IEEE 4th International Conference on Civil Aviation Safety and Information Technology (ICCASIT), Dali, China, 12-14 October 2022; pp. 1349-1354.

[6] Gao, S.; Shi, S.; Zhang, Y. Rolling Bearing Compound Fault Diagnosis Based on Parameter Optimization MCKD and Convolutional Neural Network. IEEE T. Instrum. Meas. 2022, 71, 1-8.

[7] Zhang, C.; Xu, L.; Li, X.; Wang, H. A Method of Fault Diagnosis for Rotary Equipment Based on Deep Learning. In Proceedings of the 2018 Prognostics and System Health Management Conference (PHM-Chongqing), Chongqing, China, 2018; pp. 958-962.

Comment 2. PCA and wavelet analysis for feature extraction and noise reduction could be discussed and compared e.g. Ali Sophian, etc., A feature extraction technique based on principal component analysis for pulsed Eddy current NDT, NDT & E International, Volume 36, Issue 1, 2003, Pages 37-41, ISSN 0963-8695, https://doi.org/10.1016/S0963-8695(02)00069-5.

Response: Thank you very much for the comment. We have added methods related to PCA and wavelet analysis in the introduction for fault feature extraction and noise reduction. Firstly, we analyzed the example you provided and proposed a PCA based non-destructive testing feature extraction method, which extracts relevant features through dimensionality reduction to achieve fault classification and diagnosis. Then, a literature combining PCA and wavelet analysis methods was added. Firstly, the signal was subjected to wavelet threshold denoising, and then the main feature information was obtained through PCA dimensionality reduction, which was used for subsequent fault diagnosis. Finally, it was pointed out that compared with linear prediction methods, PCA suffers from feature information loss in feature extraction, and the determination of wavelet basis functions and wavelet thresholds in wavelet analysis is a challenge. Linear prediction technology directly predicts the original signal, retaining complete feature information, and the FLP method proposed in this paper has convenient parameter optimization. The comparison between the two methods highlights the advantages of the proposed method. The added references are as follows:

[1] Sophian, A.; Tian, G.Y.; Taylor, D.; Rudlin, J. A feature extraction technique based on principal component analysis for pulsed Eddy current NDT. Ndt&E. Int. 2003, 36, 37-41.

[2] Xu, Q.; Chen, J.; Wu, B. Preprocessing of Massive Flight Data Based on Noise and Dimension Reduction. In Proceedings of the 2020 IEEE 6th International Conference on Computer and Communications (ICCC), Chengdu, China, 2020; pp. 1706-1710.

[3] Sun, Y.; Xu, A.D.; Wang, K.; Han, X.J.; Guo, H.F.; Zhao, W. A novel bearing fault diagnosis method based on principal component analysis and BP neural network. In Proceedings of the 2019 14th IEEE International Conference on Electronic Measurement & Instruments (ICEMI), Changsha, China, 2019; pp. 1125-1131.

Comment 3. Mind technical contribution and novelty.

Response: Thank you very much for the comment. We highlight the technical contribution and novelty in the abstract, introduction, and conclusion, highlighting the innovation of FLP method in the field of bearing fault diagnosis. FLP has been applied in previous studies for modeling and compression of speech signals due to its good predictive performance. Using this predictability as a distinguishing feature eliminates the need to understand pure noise data under different conditions during the noise modeling process. Modeling predictable components as noise to eliminate noise and highlight fault characteristics, ultimately achieving the goal of fault diagnosis. In theory, FLP has advantages over LP in parameter optimization and comparable predictive performance. Considering the application of LP method in the field of bearing fault diagnosis and its ability to achieve certain diagnostic results, FLP method is also applied in the field of bearing fault diagnosis. Through experiments, it is found that FLP has better fault feature extraction effect, more obvious noise suppression, and better predictive performance, which is innovative.

Comment 4. The conclusion and future work should be enhanced.

Response: Thank you very much for the comment. We have supplemented the conclusion by providing a more detailed explanation of the advantages and principles of this method, highlighting the advantages of the FLP proposed in this paper compared to other methods, and emphasizing the innovation points of this paper. And supplemented the future work, the future work is to apply this current noise elimination method FLP to other motor bearings to verify its generalization and whether it can also achieve good bearing fault diagnosis results. And whether the parameters of the FLP model in this article are also used in different motor bearings. These works require further research in the future.

Round 2

Reviewer 2 Report

Comments and Suggestions for Authors

The improvement is reasonably good. Mind further English proofread.

e.g. 'pulse Eddy current (PEC)', 'These works require further research in the future.' need English proofread.

Comments on the Quality of English Language

E.g. 'pulse Eddy current (PEC)', 'These works require further research in the future.' need English proofread.

Author Response

Comments to the Author

The improvement is reasonably good. Mind further English proofread.

Comment 1. e.g. 'pulse Eddy current (PEC)', 'These works require further research in the future.' need English proofread.

Response: Thank you very much for the comment. We are very sorry that there are still multiple grammar errors and sentence structure issues in the manuscript. We carefully proofread the manuscript several times and corrected grammar errors and sentences that reduced readability throughout the entire manuscript. Based on the example errors in the comments, correct them as follows:

  1. pulse Eddy current (PEC) --> pulse eddy current (PEC)
  2. These works require further research in the future. --> Further research is required for these works.

The remaining modifications are highlighted in blue in the manuscript, mainly aimed at improving the readability of certain sentence structures, inappropriate wording, the use of definite articles, and incorrect sentence structures.
